# Assessing the Penetrance of Dementia Services

**DOI:** 10.3390/healthcare7030081

**Published:** 2019-06-27

**Authors:** Michael Rozsa, Leon Flicker

**Affiliations:** 1Resident Medical Officer Royal Perth Hospital, Perth 6001, Australia; 2Geriatric Medicine, Western Australian Centre for Health & Ageing, Medical School, University of Western Australia, Crawley 6009, Australia; 3Geriatrician, Royal Perth Hospital, Perth 6001, Australia

**Keywords:** dementia, cognitive impairment, memory clinics, assessment penetrance

## Abstract

Introduction: This scoping review aims to identify studies that assess dementia case finding programs (such as memory clinics) on a population basis and specifically assess the impacts that these services have on the diagnosis and management of dementia within a specific population. Methods: We conducted a literature review using the PubMed database, Ovid search engine, and records identified from external sources. This review assessed studies that contained data on patients diagnosed with dementia within a set population, reviewed the impact of specialty services on the diagnosis and management of such patients, and evaluated how this compared to data estimates for that population catchment. Results: The literature review yielded 1106 unique studies, of which only five were determined to be relevant based on the inclusion criteria. There was considerable variation between the primary outcome measures of the five studies included, and a quantitative meta-analysis could not be performed. Discussion: There are currently limited data on the fraction of the total population of people with dementia that are diagnosed and managed by specialised dementia assessment services within a set population. Further studies investigating how these services impact the incidence and prevalence of dementia diagnosis and ongoing management are required.

## 1. Introduction

Dementia is a condition that is becoming more prevalent worldwide due to population ageing [1,2,3,4], and is placing an increasing burden on the resources within healthcare systems [1,2,3,4,5,6]. The early detection and management of dementia can help to ease this burden through the timely implementation of appropriate management, and by allowing for arrangements to be made to prolong independence and quality of life before the condition has advanced to a stage where supports are more difficult to implement [2,5,7].

Dementia diagnosis and management can be challenging because of disease factors, as well as patient factors [8,9]. The complicating disease factors include the gradual onset of disease [7,10], the shortage of time and/or knowledge in primary care to recognise or investigate cognitive impairment [8,10,11,12,13], the lack of cost-effective and accessible biomarkers indicating disease stage and progression [14], and the absence of one unifying cognitive assessment tool for patients from different cultures, socioeconomic backgrounds, or education levels [15,16,17]. A further complicating factor is the diagnostic uncertainty created due to the often mixed or multifactorial pathologies that can contribute to dementia [7,8,9].

Patient factors contributing to diagnostic difficulty include poor insight into their own cognitive decline, fear of or unwillingness to accept a diagnosis of dementia, and logistical issues with review by an outpatient service that can diagnose dementia due to poor mobility and reduced independence [18,19,20,21].

Because of these factors, significant morbidity can arise from delayed or inadequate diagnosis leading to suboptimal management of patients with cognitive impairment and dementia [8,9,10]. Dementia assessment services should be evaluated on their accuracy of assessment and effectiveness in linking patients to appropriate support services. However, another important aspect of such services is the overall access and penetration of services for a population.

Many healthcare providers and services are involved in identifying and providing care for people with dementia. We have chosen to assess the penetrance of memory assessment services in order to evaluate the role they play in the complex multidisciplinary process of diagnosing and managing dementia. As is recommended by the national guidelines in Australia, all patients suspected to have a diagnosis of dementia should be offered assessment by memory assessment services [22]. Memory assessment services vary significantly in structure and function globally, and by exploring the role of these services we intend to elucidate to what degree these services assess and manage their respective populations of patients with dementia. 

In this scoping review, we aim to identify studies that assess dementia case finding programs (such as memory clinics) on a population basis and specifically assess the impacts that these services have on the diagnosis and management of dementia within that population. Additionally, we aim to evaluate what barriers to these specialised diagnostic and management services may exist, and how these barriers affect diagnosis and management. 

Based on these studies, we hope to compare the estimated incident diagnoses of dementia within a catchment area (based on population data) with the actual number of dementia cases identified and/or receiving further management of cognitive impairment within that catchment across a range of international populations. This is with the further goal of contributing to our understanding of the impact that services that diagnose and manage dementia have on a proportion of the estimated population with dementia. 

## 2. Materials and Methods

In this scoping review, we aimed to review observational studies that assessed the number of cases of dementia in a set population or catchment, and how the clinics or teams designed to specifically assess for and manage dementia affected the rates of dementia diagnosis and treatment on an ongoing basis.

We conducted a PubMed search with the following terms:

(Dementia/diagnosis[MAJR] OR “dementia/epidemiology” [MAJR]) AND (“mass screening” [MeSH Terms] OR “geriatric assessment” [MeSH Terms]) AND (“predictive value of tests” [MeSH Terms] OR “sensitivity and specificity” [MeSH Terms] OR burden [All Fields] OR prevalent [All Fields] OR “prevalence” [All Fields] OR “prevalence” [MeSH Terms] OR incident[All Fields] OR “incidence” [All Fields] OR “incidence” [MeSH Terms] OR proportion [All Fields] OR undiagnosed [All Fields] OR missed [All Fields]).

We also conducted a search using the Ovid search engine, which reviews Medline, Embase, and PsychInfo databases, with the following terms:

(exp *dementia/di or exp *dementia/ep) and (exp Mass Screening/or exp geriatric assessment/) and ((Burden or Prevalen* or Inciden* or Proportio* or Undiagnosed or Missed).mp or exp predictive value of tests/or exp “sensitivity and specificity”/). 

To identify suitable studies, we initially reviewed the abstracts, and if in doubt the full papers were retrieved. We used the following inclusion criteria: studies that identified diagnoses or management of dementia within a set population (yes/no), whether patients were reviewed by specialised services for diagnosing and/or managing dementia (yes/no), were numerical data regarding the diagnosis and/or management of dementia recorded as part of the study (yes/no), was there a comparison to estimated dementia prevalence within the assessed population (yes/no), and was an assessment made as to the impact that the specialised services had on the proportion of patients who received a diagnosis of dementia (yes/no). We limited our studies to those published in the English language with the full-text publication available for access.

## 3. Results

The literature search performed using the PubMed database, Ovid search engine, and records identified from external sources yielded 2064 studies for possible inclusion (1106 after the removal of duplicate studies found across the data sources). These searches were current up to 6 February 2019. After review, only five papers were found to be suitable for inclusion as per the eligibility and inclusion criteria. Figure 1 shows a flow chart of the papers assessed for this study. A summary of the key parameters and findings from the five studies used is presented in Table 1. There were no studies published relevant to the search terms prior to 1989.

Of the 1106 records reviewed (with duplicates excluded from the total of 2064 records found) from the three data sources, 224 records were included for further review of the abstracts, with the other 882 records being excluded due to clear differences between what those papers assessed and our inclusion criteria. Of these 224 studies, 36 were selected for full-text review, as the other 188 did not include numerical data related to dementia diagnosis or management and/or explore the role of specialised services in dementia diagnosis and management. Of the 36 full-text studies reviewed, only five were considered to meet the inclusion criteria. The other 31 papers did not compare the actual proportions of diagnosis or management of dementia patients within a catchment with a data estimate, or did not explore the impact that specialty dementia services have on the proportion of dementia patients receiving appropriate assessment (see Figure 1).

A study from Jitapunkal et al. demonstrated a high prevalence of undiagnosed dementia in a developing nation, where specialist services for dementia were relatively scant [23]. 

The effect of access to specialist dementia services was noted by the inverse relationship between the incidence of a diagnosis of dementia, and the geographical distance between a patient’s usual abode and their nearest specialised dementia service or clinic (Jørgensen et al.) [24].

The McCarten et al. study demonstrated increased detection of dementia when a dementia specific service was included as part of the routine medical management for elderly veterans. There was an approximate threefold (11% compared to 4%) increase in the detection of cognitive impairment in patients who were reviewed by this service in comparison to those who were reviewed through conventional pathways without the specialised dementia service [25].

The SveDem study provides a large sample of patients with dementia who were recorded in a national register. This study estimates that over a third of all new cases of dementia were recorded in this register, and that specialty services provided between one half to three-quarters of the diagnosed cases entered into the register [26]. 

Our own study, Rozsa et al., was the only study that provided information regarding an estimated fraction of dementia patients who were diagnosed or managed by specialised dementia services within a set catchment, and estimated that 28% of incident cases were assessed by publicly available memory specialist services, and that 19% of prevalent dementia cases were assessed by the ACAT (aged care assessment team) in a calendar year [27]. 

## 4. Discussion

The aim of this scoping review was to analyse studies that aim to determine the penetration of routine clinical services for overall numbers of people with dementia within a population. Only our own study provided this comparison. This study assessed a constrained sample of less than 10% of the population of a moderately sized Australian city. This result is surprising, as such basic data are essential in determining whether access to services is adequate and whether changes to services have the desired effect of increasing the penetrance of specialist dementia services within a general population. 

Significant variation between the primary outcomes and measures existed between the studies included in this review, but the primary findings obtained from these studies were similar. Firstly, in all populations in all countries there were a large number of people who had undiagnosed cognitive impairment/dementia. These are probably the majority of the people with cognitive impairment or dementia. Secondly, specialised diagnostic and management services, such as memory clinics, improved the rates of diagnosis and ongoing follow-up care for these patients. Subsequently, reduced access to these clinics (through geographical, socioeconomic, or other factors) limited the number of patients receiving the appropriate diagnosis and management. 

The most significant finding from this scoping review is not the information obtained from the selected studies themselves, but rather the lack of studies relevant to the inclusion criteria that look at the impact of memory clinics and services on the rates of dementia diagnosis and management. As a result, it is still difficult to accurately ascertain to what degree these services affect the prevalence and incidence of cognitive impairment diagnosis and ongoing management. This is despite the available data which suggests that these services may improve the rate of diagnosis by providing a formal framework for these patients to receive ongoing care and referral for issues related to cognitive impairment. 

It can be argued that the value of specialized dementia assessment services has not been established, and thus the penetrance of these services into the overall population is not important. We would contend that, as we are at a stage where different service models are available worldwide, including low and middle income countries, now is an opportune time to compare these service models based on their overall ability to deal with their individual populations. It may be that these services will never be available to the majority or people with dementia and that other, more general, services will be the predominant model. This issue will be clarified by further studies in this area. 

This study does have some limitations. Only studies published in the English language with a full-text available were eligible for inclusion in this study and subsequently assessed. The limited number of studies and their disparate nature precluded any form of quantitative analysis. 

## 5. Conclusions

From this scoping review, we can conclude that there is currently limited data available on the fraction of the total population of people with dementia that are diagnosed and managed by specialised dementia assessment services. This in turn limits our understanding of the impact that such services may have on the proportion of all dementia cases that are identified, diagnosed and managed within a set population. Further studies investigating how these services impact the incidence and prevalence of dementia diagnosis and ongoing management is required to better understand how to best reduce population morbidity, carer stress, and the burden on the healthcare system that is caused by dementia.

## Figures and Tables

**Figure 1 healthcare-07-00081-f001:**
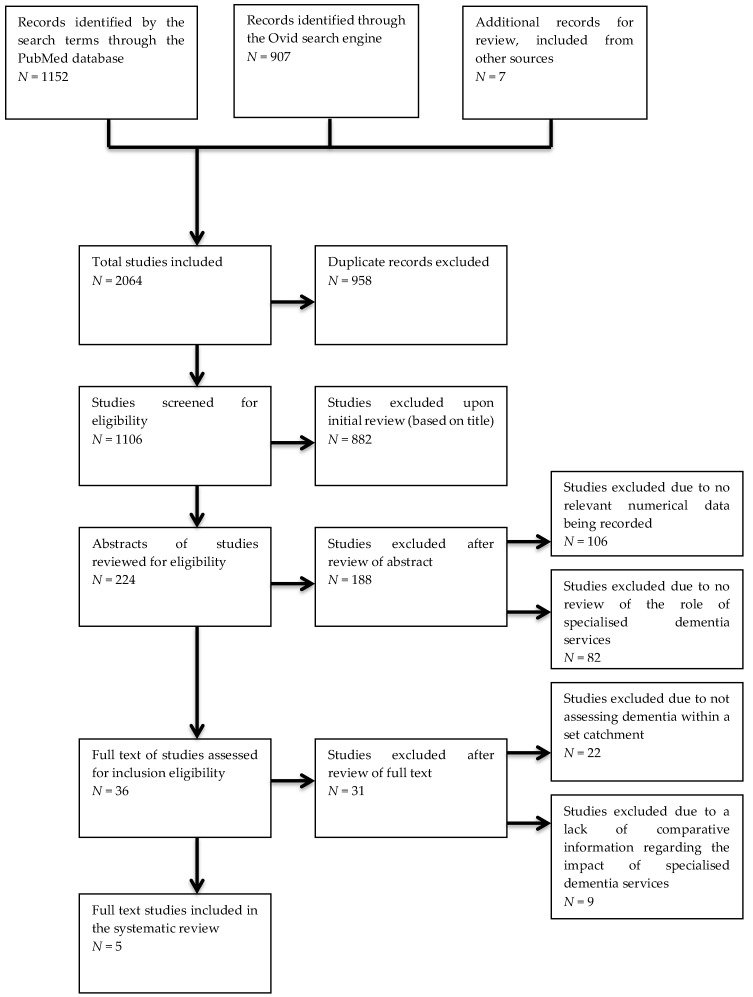
Flow diagram of the studies identified and assessed through the eligibility and inclusion criteria.

**Table 1 healthcare-07-00081-t001:** Description of the included studies.

Studies Included	Sample Assessed	Sample Size	Diagnostic Tools	Findings
Jitapunkul et al.: Un-diagnosed dementia and the value of serial cognitive impairment screening in the developing world [23]	Participants from the CERB study of the Romklao community in Bangkok, Thailand aged 60 and over (who were able to be contacted as part of the study).	420	Chula Mental Test (CMT)	Of the 420 participants, 23 were diagnosed with dementia, with only one (4.4%) of the 23 having a diagnosis of dementia prior to this assessment. There was no previous diagnosis of dementia for 95.6% of the patients, with the prevalence of undiagnosed dementia being 5.3% (95%, CI 4.1–6.3%) within the total assessed population.
Jørgensen et al.: Time trend in Alzheimer diagnosis and the association between distance to Alzheimer clinic and Alzheimer diagnosis [24]	All individuals aged 65 years and over living in Denmark from 1/1/2000 until diagnosis, death, emigration or the end point on 31/12/2009.	830,869	No documentation regarding the diagnostic tools used. Patients were included either through an Alzheimer’s diagnosis, as per ICD-10, or by the first prescription of medication for Alzheimer’s.	When stratified by the distance to dementia assessment centres, when compared to people living <20 km away, people living 20–39 km from assessment centres had a hazard ratio (HR) of 0.80 (CI 0.70–0.92) for a dementia diagnosis. Those living 40–59 km away had a HR of 0.65 (CI 0.52–0.81). Those living 60+ km away had a HR of 0.81 (CI 0.59–1.11). The inverse relationship between the geographical distance from patients to memory clinics and the rate of diagnosis of dementia indicates that reduced geographical access to services providing an accurate diagnosis of dementia results in less people receiving the appropriate diagnosis and treatment.
McCarten et al.: Finding dementia in primary care: The results of a clinical demonstration project [25]	Veterans across 8 medical centres in Virginia, USA who were aged 70 or older, in generally stable health, able to complete the screening, and did not have a prior diagnosis of cognitive impairment.	8063	The Mini-Cog as an initial test, with further tests including the MoCA, NPI-Q, the CPT, the dependence screen, a driving screen, and a caregiver needs assessment.	A specialised dementia screening program for veterans allowed for the diagnosis of newly recognised cognitive impairment in 11% (902/8063) of cases, in comparison to 4% of cases (1242/28349) of newly diagnosed cognitive impairment in similar veteran clinics without a dementia screening program.
Religa et al.: SveDem, the Swedish Dementia Registry—A tool for improving the quality of diagnostics, treatment and care of dementia patients in clinical practice [26]	All patients with a new diagnosis of dementia made in Sweden between 1/5/2007 and 31/12/2012 that was reported to the SveDem national dementia registry.	28722 subjects registered with new diagnoses in Sweden between 1/5/2007 and 31/12/2012.	Multiple, including a baseline MMSE and clock drawing test (as well as various other modalities), as well as organic screening via imaging and blood tests.	Of the new dementia diagnoses made from 2010 to 2012 (once specialty and primary care services were aware of the registry), an estimated 36% of the 20,000 potential incident dementia diagnoses in Sweden each year were included in Svedem. Specialty services were responsible for more of the new diagnoses registered into SveDem than primary care each year, but the gap between the diagnostic rates by these services was reduced towards the conclusion of the data collection (2010: 73.8% vs. 26.2%; 2011: 64.9% vs. 35.1%; 2012: 51.9% vs. 48.1%).
Rozsa et al.: Assessing people with dementia: The role of the aged care assessment team (ACAT) and memory clinics [27]	All patients reviewed by the ACAT team and by the memory clinics of the public healthcare system for an inner city catchment area of Perth, Western Australia, in the 2012 calendar year.	1005 subjects reviewed by the ACAT, 186 subjects reviewed by memory clinics.	Multiple (including MMSE, Cambridge cognitive assessment, GDS, Katz Index of ADLs, and neuro-psychiatric inventory).	Of an estimated 1260 prevalent dementia cases within an inner city population over the age of 70, 19% (*n* = 241) were assessed by an aged care assessment team service within a calendar year (from the 1005 patients reviewed that year). Of an estimated 286 incident dementia cases from the same population, 28% (*n* = 82) were diagnosed at a public memory clinic servicing that catchment from the 186 new referrals reviewed by the clinic in a calendar year.

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
