# Peer review of "Assessing the Penetrance of Dementia Services"

_healthcare, 2019, doi:10.3390/healthcare7030081_

Round 1

Reviewer 1 Report

The authors have addressed the concerns raised by the reviewer. Thank you for the attention to this. 

Author Response

Thank you for your review of the study and recommendations made to improve this manuscript

Reviewer 2 Report

Thank you for the opportunity to review the revision to the manuscript "Assessing the Penetrance of Dementia Services". Although the authors have a addressed some of the suggestions from my initial review, I feel additional revisions are needed for this manuscript to be a valuable addition to the literature.

1. The authors do not make the case for the role of dementia specialty clinics in the diagnosis and treatment of dementia. They reference memory clinics as being a method of "case identification" but since memory clinics do not perform population based screening, I do not see how they are a method for case identification. Rather, memory clinics are generally focused on the evaluation and treatment of patients who are referred due to memory complaint.

2. There is still limited synthesis of the studies included in for review. There is no discussion of what the studies have in common, how they are different, and what broad conclusions can be made based on considering all the studies together.

3.The authors should consider inclusion of the systematic review in their introduction to provide additional background on the need for increased diagnosis.

Lang, Linda & Clifford, Angela & Wei, Li & Zhang, Dongmei & Leung, Daryl & Augustine, Glenda & Danat, Isaac & Zhou, Weiju & Copeland, John & Anstey, Kaarin & Chen, Ruoling. (2017). Prevalence and determinants of undetected dementia in the community: A systematic literature review and a meta-analysis. BMJ Open. 7. e011146. 10.1136/bmjopen-2016-0111

Author Response

Thank you this re-review. We have addressed all the comments made.

We have accepted all previous changes in the document and made the following changes as described below.

1. The authors do not make the case for the role of dementia specialty clinics in the diagnosis and treatment of dementia. They reference memory clinics as being a method of "case identification" but since memory clinics do not perform population based screening, I do not see how they are a method for case identification. Rather, memory clinics are generally focused on the evaluation and treatment of patients who are referred due to memory complaint.

We apologise that we have not made it clear enough to the reviewer. There is no place in the article where we have suggested that memory clinics or other speciality clinics are involved in any sort of population screening. What we are investigating is what proportion of people with dementia within a defined population have been seen by this type of clinic. Whether it should be 1% or 99% is unknown, but the comparison between different systems is of interest. We are seeking studies of “penetrance” of this type of clinic within a population. To make this clearer, we have added the word “penetrance” to the abstract Line 11 and to the introduction, line 62.

2. There is still limited synthesis of the studies included in for review. There is no discussion of what the studies have in common, how they are different, and what broad conclusions can be made based on considering all the studies together.

We agree that the studies have little in common but that is one of the major findings from this review. We have highlighted how different the studies are, despite these studies being closest in satisfying the predetermined inclusion criteria that we sent to the previous Editors of this special issue. There is a need for this very basic type of health services research in determining the usage of specialised clinics.

3.The authors should consider inclusion of the systematic review in their introduction to provide additional background on the need for increased diagnosis.

Lang, Linda & Clifford, Angela & Wei, Li & Zhang, Dongmei & Leung, Daryl & Augustine, Glenda & Danat, Isaac & Zhou, Weiju & Copeland, John & Anstey, Kaarin & Chen, Ruoling. (2017). Prevalence and determinants of undetected dementia in the community: A systematic literature review and a meta-analysis. BMJ Open. 7. e011146. 10.1136/bmjopen-2016-0111

We agree that this paper would be useful and have added it to the introduction. We have included the following in lines 44 to 47

“For these, and many other reasons, the prevalence of undetected dementia in various middle and high income countries, in community and residential care populations, was found to be 61.7%, and ranged in individual studies from 31% to 96% (22).” 

We have renumbered the references 22-28 accordingly.

We trust that these revisions now satisfy the residual comments from the reviewer and look forward to this article being included in the special issue for which it was solicited.

Round 2

Reviewer 2 Report

No additional comments

This manuscript is a resubmission of an earlier submission. The following is a list of the peer review reports and author responses from that submission.

Round 1

Reviewer 1 Report

Thank you for the opportunity to review this paper. It is well written and the topic is interesting, but it included only five papers, all of which were very different. Thus, this is at most a scoping review.  It is not a systematic review and instead meets the definition of Grant and Booth (2009), which is "preliminary assessment of potential size and scope of available research literature.  Aims to identify nature and extent of research evidence (usually including ongoing research)." With this in mind, the authors should have structured the review differently, and instead of simply listing a set of inclusion/exclusion criteria for the review, they should have carefully framed a research question, i.e. ‘what is the nature and outcome of dementia case finding initiatives’ (or something like that).

My main issues with the paper is that it purports to assess the value of 'memory clinics' based on their ‘penetrance’ of services for a population’ – in other words, the degree to which they engage in case finding. I really struggle with this premise. With the rising prevalence of dementia globally, it is becoming increasingly clear that Memory Assessment Services (‘MAS’) cannot possible take on the burden of all assessment and diagnostic tasks related to dementia. This model is simply unsustainable. In several centres in Europe, MAS are now dedicated to assessing and managing only complex cases. In the UK, several MAS will only assess and diagnose after detailed initial work-ups have been undertaken at the primary care level. For MAS that don’t do this, they have nurse-led clinics with doctors only being asked to consult on atypical or complex cases. In other words, the role of MAS as ‘case finder’ and having deep penetrance, is likely flawed in this age of resource constraints and growing demands of an ageing population.  Furthermore, case finding is not one of the evaluation criteria of the few MAS accreditation schemes that exist. 

Thus, since the role of the vast majority of MAS is different to what the authors propose, it is not surprising that only 5 out of over 2000 studies met their inclusion criteria.  In fact, they authors point out that their study ‘was the only study that provided information regarding an estimated fraction of dementia patients who were diagnosed or managed by specialised dementia services within a set catchment’  this suggests that no one else had the same objective!

The studies included by the authors (including one of their own) appear to have diverse objectives. A few of them appear to be specific case finding, outreach programmes. These are not ‘memory assessment services’ in the traditional sense. The Thai paper appears to be a research study developing a cohort or ascertaining prevalence for the purpose of epidemiology. Hence, I am not sure the review is doing the topic justice as the goals differ across studies. 

With these comments in mind, to make sense of the topic of the review, the authors should move away from the notion of ‘memory assessment clinics’ and their activities (as their introductory paragraphs describe) and re-frame the background/intro to focus on screening programmes or case-finding programmes, perhaps by bringing in examples from other disease areas. Again, screening and case finding is not traditionally the role of MAS. Alternatively, discussing the role of primary care in screening for dementia (a very contentious issue in some countries) could be the focus.

Author Response

Coverletter to reviewer 1:

Comments 1: Thank you for the opportunity to review this paper. It is well written and the topic is interesting, but it included only five papers, all of which were very different. Thus, this is at most a scoping review. It is not a systematic review and instead meets the definition of Grant and Booth (2009), which is "preliminary assessment of potential size and scope of available research literature. Aims to identify nature and extent of research evidence (usually including ongoing research)." With this in mind, the authors should have structured the review differently, and instead of simply listing a set of inclusion/exclusion criteria for the review, they should have carefully framed a research question, i.e. ‘what is the nature and outcome of dementia case finding initiatives’ (or something like that).

Authors’ responses: We initially planned and performed this study as a systematic review with previously determined inclusion criteria. However the reviewer is correct in that as we searched the literature the studies found did not directly address our inclusion criteria, and they manifested a wide variety of aims and objectives. We have now read a recent review from Pham and colleagues entitled “A scoping review of scoping reviews: advancing the approach and enhancing the consistency” (Res Synth Methods. 2014; 5:371-85) and will accept that our review better suits the title of scoping review as opposed to systematic review, although of course the definition of scoping reviews is by no means clear cut. We have thus changed all references to the study design as a scoping review (See Page lines 9, 61, 73, 152 [within the flowchart], 161, 175, and 193).

Comments 2: My main issues with the paper is that it purports to assess the value of 'memory clinics' based on their ‘penetrance’ of services for a population’ – in other words, the degree to which they engage in case finding. I really struggle with this premise. With the rising prevalence of dementia globally, it is becoming increasingly clear that Memory Assessment Services (‘MAS’) cannot possible take on the burden of all assessment and diagnostic tasks related to dementia. This model is simply unsustainable. In several centres in Europe, MAS are now dedicated to assessing and managing only complex cases. In the UK, several MAS will only assess and diagnose after detailed initial work-ups have been undertaken at the primary care level. For MAS that don’t do this, they have nurse-led clinics with doctors only being asked to consult on atypical or complex cases. In other words, the role of MAS as ‘case finder’ and having deep penetrance, is likely flawed in this age of resource constraints and growing demands of an ageing population. Furthermore, case finding is not one of the evaluation criteria of the few MAS accreditation schemes that exist. Thus, since the role of the vast majority of MAS is different to what the authors propose, it is not surprising that only 5 out of over 2000 studies met their inclusion criteria. In fact, they authors point out that their study ‘was the only study that provided information regarding an estimated fraction of dementia patients who were diagnosed or managed by specialised dementia services within a set catchment’ this suggests that no one else had the same objective!

Authors’ responses: The fact that penetrance is not one of the current criteria for evaluation of Memory Assessment Services, does not negate the importance, or even interest, in this outcome. Even accepting that most patients will not be assessed by specialised Memory Assessment Services it is still of interest in comparing across countries and service models as the reviewer has just demonstrated. We have not maintained in this article that all patients should be seen by Memory assessment services, but a recommendation to offer patients assessment by such services has been made by the Australian National Health and Medical Research Committee (NHMRC) Clinical Practice Guidelines and Principles of Care for People with Dementia. Sydney. Guideline Adaptation Committee; 2016. NHMRC. We have reflected this with an additional paragraph in the introduction (located from line 53-60).

Comments 3: The studies included by the authors (including one of their own) appear to have diverse objectives. A few of them appear to be specific case finding, outreach programmes. These are not ‘memory assessment services’ in the traditional sense. The Thai paper appears to be a research study developing a cohort or ascertaining prevalence for the purpose of epidemiology. Hence, I am not sure the review is doing the topic justice as the goals differ across studies.

Authors’ responses: Yes we accept that these studies are all quite different and could best be part of a scoping review rather than a systematic review as first intended.

Comments 4: With these comments in mind, to make sense of the topic of the review, the authors should move away from the notion of ‘memory assessment clinics’ and their activities (as their introductory paragraphs describe) and re-frame the background/intro to focus on screening programmes or case-finding programmes, perhaps by bringing in examples from other disease areas. Again, screening and case finding is not traditionally the role of MAS. Alternatively, discussing the role of primary care in screening for dementia (a very contentious issue in some countries) could be the focus.

Authors’ responses: The role of primary care in the assessment of dementia is a very interesting one but is not the purpose of this paper. We have specifically tried to elucidate a unique aspect of specialised memory assessment services and highlighted a lack of information within this area. The former editors of this specific issue initially contacted us about our previous paper (Rozsa, Ford, Flicker. Intern. Med. Journal. 2016), from which we subsequently proposed a systematic review based upon this study and the concept of assessment penetrance to the editors. This was done before we performed our search of the literature, and we then found limited and disparate studies that satisfied our pre-determined inclusion criteria.

Reviewer 2 Report

Overall this is a clear, concise and well-written manuscript. The method are appropriate, although this is not so much relevant for geriatric mental health.  

Good Luck 

Author Response

Overall this is a clear, concise and well-written manuscript. The method are appropriate, although this is not so much relevant for geriatric mental health. 

Good Luck

Thank you for your supportive comments. However, we maintain that evaluating and better understanding the role that memory assessment services provide in the care of patients with dementia will help to inform how services provision can be improved to these patients, hopefully to improve their general well being.

Reviewer 3 Report

I appreciate the opportunity to review this abstract titled "Assessing the Penetrance of Dementia Services".  This is an important topic that has significance to multiple medical specialties.  The officers have done quite a lot of work with regard to the design and cert strategy.  However the paper is lacking in its review of the relevant studies as well as being able to since the size this information into a strong review article.  Please see recommendations for improvement below

  Abstract

 the abstract should include the number of unique articles rather than including the duplicates

 introduction

 the introduction should be expanded to provide more detail about the rationale for under taking this review.  It is not clear from the review why this is important topic.  Specifically it is unclear why the author's have chosen such specific criteria for inclusion/exclusion studies involving dementia Care Services.

 it was somewhat unclear whether the others are looking to understand more about screening for dementia, case identification for dementia or diagnostic assessment for dementia.  Specialty dementia care programs may or may not have interventions that improved case identification in their community but rather may focus simply on confirming diagnosis after a primary care physician has identified a concern.

 Results

 in the flow diagram, it would be helpful to detail how many studies were excluded for specific reasons rather than grouping them altogether

 a more detailed description of the studies and their specific findings is needed.

 there should be more synthesis regarding common themes among studies

 discussion

 the discussion is quite limited and does not adequately synthesizing information provided in the review.  With the current discussion it is still somewhat unclear what is the relevance of this review article and water the main take away points for the reader.

 There is a concern that the scope of the review may have been overly narrow, leaving out many studies that address dementia specialty care and dementia prevalence, but with exactly the inclusion criteria determined by the authors

 it would be important to the it would be important to address the role of collaborative/integrated Behavioral Healthcare as another mode of screening, diagnosis and treatment of dementia.  This would be a different approach to population health management of dementia within primary care and does not rely on dementia specialty programs.

Author Response

Coverletter to reviewer 3:

I appreciate the opportunity to review this abstract titled "Assessing the Penetrance of Dementia Services". This is an important topic that has significance to multiple medical specialties. The officers have done quite a lot of work with regard to the design and cert strategy. However the paper is lacking in its review of the relevant studies as well as being able to since the size this information into a strong review article. Please see recommendations for improvement below.

Comments 1: Abstract

The abstract should include the number of unique articles rather than including the duplicates

Authors’ response: We have altered the abstract to now only include the 1106 unique articles (see line 17).

Comments 2: Introduction

The introduction should be expanded to provide more detail about the rationale for under taking this review. It is not clear from the review why this is important topic. Specifically it is unclear why the author's have chosen such specific criteria for inclusion/exclusion studies involving dementia care services.

It was somewhat unclear whether the others are looking to understand more about screening for dementia, case identification for dementia or diagnostic assessment for dementia.

Specialty dementia care programs may or may not have interventions that improved case identification in their community but rather may focus simply on confirming diagnosis after a primary care physician has identified a concern.

Authors’ response: We have expanded the introduction to include more information as to why this is an important subject (following the suggestions of Reviewer 1) and why the penetrance of Memory Assessment Services is influenced by case finding programs (See lines 53-60).

Comments 3: Results

In the flow diagram, it would be helpful to detail how many studies were excluded for specific reasons rather than grouping them altogether

A more detailed description of the studies and their specific findings is needed. There should be more synthesis regarding common themes among studies

Authors’ response: We have added some additional details to the flow chart indicating the specific reasons for exclusion, but did not want the chart to be so detailed as to be illegible (See the flowchart on line 152)

Comments 4: Discussion

The discussion is quite limited and does not adequately synthesizing information provided in the review. With the current discussion it is still somewhat unclear what is the relevance of this review article and water the main take away points for the reader.

There is a concern that the scope of the review may have been overly narrow, leaving out many studies that address dementia specialty care and dementia prevalence, but with exactly the inclusion criteria determined by the authors

Authors’ response: We have now attempted to more fully synthesise the information available but the studies that most closely fulfilled the inclusion criteria were from disparate areas.

Comments 5: It would be important to the it would be important to address the role of collaborative/integrated Behavioral Healthcare as another mode of screening, diagnosis and treatment of dementia. This would be a different approach to population health management of dementia within primary care and does not rely on dementia specialty programs.

Authors’ response: Yes these programs may hold promise in the care of people with dementia but unfortunately programs of behavioural health care were not detected by our rather broad search strategy based on our predetermined inclusion criteria. They could be usefully included in a separate review.